# Profiling Experiences of Bullying in the Elementary School: The Role of Gender

**DOI:** 10.3390/children10040610

**Published:** 2023-03-23

**Authors:** Georgios Sideridis, Mohammed H. Alghamdi

**Affiliations:** 1Boston Children’s Hospital, Harvard Medical School, Boston, MA 02131, USA; 2Department of Primary Education, National and Kapodistrian University of Athens, Navarinou 13A, 10680 Athens, Greece; 3Department of Self Development Skills, King Saud University, P.O. Box 2454, Riyadh 11451, Saudi Arabia; mhalghamdi@ksu.edu.sa

**Keywords:** bullying, TIMSS, gender effects, fourth grade

## Abstract

The purpose of the present study was to profile bullying behaviors in elementary schools in Saudi Arabia. A secondary purpose was to examine differences in bullying behaviors across gender. Participants were 3867 fourth graders who completed surveys during the TIMSS 2019 survey. An 11-item bullying experience scale was utilized with good internal consistency reliability. Data were analyzed using latent class analysis with Mplus 8.9 to identify profiles of bullying experiences. The results indicated the presence of five profiles with levels of low, medium, and high bullying experiences, as well as two profiles with no cyberbullying experiences and medium high and medium low physical and verbal instances of bullying. Gender effects were highly pronounced, with most maladaptive bullying profiles being predominantly male. It is concluded that physical bullying is mainly occupied by males and the levels of cyberbullying are generally low in the elementary school grades. Implications for educational policy can clearly direct the development of support groups and expert counseling for both bullies and victims, staff training for identification and course of action, and the development of standardized school policies when such incidences occur.

## 1. Introduction

Bullying in schools has received pandemic proportions as rates have skyrocketed to 20% in recent years [1]. Based on Wikipedia: “Bullying is the use of force, coercion, hurtful teasing or threat, to abuse, aggressively dominate or intimidate” that is also habitual and repeated. Experiences of bullying have been almost universal and have been linked to emotional problems and anxiety, maladjustment, impulsivity, psychopathology, low confidence, and low academic achievement [2,3,4,5,6,7,8,9,10,11]. In more severe occasions, experiences of bullying have been linked to suicidal ideation and even death [12,13,14]. Interestingly, suicidal ideation is more prevalent in females compared with males [12]. Thus, the implications of bullying experiences are severe and require urgent attention [15]. Particularly, in Saudi Arabia, a strong negative relationship has been observed between bullying and mathematics achievement in grade 4 using the 2015 data of the TIMSS [16]. Saudi Arabia has been ranked 38th among 57 countries, with a mean of 9.5 with scores between countries ranging between 9.1 and 11.0; thus, its “standing” in bullying is below the international average, but still has noticeable levels.

While bullying rates are on the rise, researchers have attempted to elaborate on the causal mechanisms of bullying. To this end, they have identified the increased use of social media as potential causal mechanisms, making it easier to behave in that manner compared with in face-to-face social experiences [17], peer pressures [18], abusive or neglectful family environment [19], personal dispositions [20], and cultural factors [21]. 

### 1.1. Bullying and Gender Differences

Bullying as a social phenomenon reflects a significant public health concern jeopardizing school safety [22]. Despite not having a unanimous definition, bullying is characterized by repeated instances to hurt or harm a person who is unable to defend herself/himself [23]. Four typologies of bullying have been proposed, namely, physical, verbal, relational, and cyber [24].

In two large meta-analytic studies [25,26], boys were found to have adopted the role of bully significantly more compared with girls. However, despite this omnibus finding, levels of bullying across gender are moderated by the type of bullying. For example, physical bullying is more prevalent in boys [25], as well as cyberbullying [27,28,29], whereas relational bullying is more prevalent in girls [30]. With the advancement of social media, more cyber bullying is expected across both genders.

### 1.2. Importance and Goals of the Present Study

Bullying is a huge problem in schools, with increased rates over time and continuation from elementary to middle and high school to adulthood [31]. The vast majority of the empirical literature is focused on adolescence [32], thus there is less knowledge about the prevalence and patterns of bullying behaviors observed in elementary school years, which is also largely inconclusive. Thus, person-centered approaches would be useful for profiling bullying behaviors in elementary schools, and informing on existing and emerging bullying subtypes as the focus is on grouping persons rather than identifying relationships between variables. A person-based approach can account for individual differences and also inform theory and guide the development of specific interventions. Furthermore, although levels of bullying in Saudi Arabia are below the international norm are still at high levels. The patterns and levels are particularly interesting to capture in light of gender differences in academic achievement, favoring females [33], and the fact that the teaching arrangement is segregated across gender. Gender differences are furthermore unique in Saudi Arabia, as gender separation is the norm across public domains as well [34,35].

The purpose of the present study was to profile bullying behaviors in elementary schools in Saudi Arabia as a means of identifying distinct patterns of bullying behaviors. A secondary goal was to examine differences in bullying behaviors across gender. Specifically, we posited the following non-directional research questions:
What is the composition and number of latent subgroups related to bullying behaviors?Are there differences across gender in the level and type of bullying behaviors?


## 2. Method

### 2.1. Participants and Procedures

The participants were 3867 fourth graders who participated in the TIMSS 2019 and were part of the data collection in Saudi Arabia. Only participants with full data were included. There were 1692 males (43.7%) and 2182 females (56.3%). The mean age was 10.79 years (S.D. = 1.352). All students were of Arabic origin with 97.4% born in Saudi Arabia. Students came from 217 schools across all areas of Saudi Arabia. Schools were located in primarily urban areas (i.e., cities with 100,000 population or more) at a rate of 61.6%, suburban areas (with a population of 15,000 to 99,000) at a rate of 15.7%, and rural areas (less than 30 k population) at a 22.7% rate. With regard to SES, 54.9% of the schools were characterized as “more affluent” and 25.8% as “more disadvantaged” based on the criteria developed by TIMSS using data collected from principals. The remaining 19.3% of the schools were characterized as “neither more affluent nor more disadvantaged”.

Participants were selected using a stratified two-stage cluster sample design. The first stage involved sampling schools with selection probabilities proportional to their size. Stratification involved gender and school type (private/or public/international). The second stage involved selecting intact classes using a “within-school sampling” methodology developed by IEA Hamburg and Statistics Canada. The National Research Coordinators were then responsible for carrying out the assessments. Non-response bias was accounted for by developing minimum inclusionary standards for schools (at least 85% participation). One classroom per school was sampled and the coverage was 100%. Exclusionary criteria involved special education schools, very small schools (n < 6), and schools in Jizan Najran and part of Asir. Furthermore, non-native language speakers were excluded. More information on sampling and representation can be found in the original source (https://timssandpirls.bc.edu/timss2019/ (accessed on 15 January 2023)).

The National School Coordinators arranged the dates, times, and places of testing. They coordinated all procedures by collecting parent permission forms where necessary, the production and distribution of questionnaires, and completing tracking forms. They were also responsible for securing assessment material and also arranging the return of completed forms to the national center post-administration.

### 2.2. Measure: Bullying Scale of the TIMSS

An 11-item construct of bullying behaviors as assessed on TIMSS was utilized in the present study (see Table 1). Internal consistency reliability was assessed using omega which is appropriate for congeneric measures, and the popular Cronbach’s alpha. Omega was 0.871 and alpha was 0.863, both being acceptable. The items were as follows: (1) made fun of me or called me names; (2) left me out of their games or activities; (3) spread lies about me; (4) stole something from me; (5) damaged something of mine on purpose; (6) hit or hurt me; (7) made me do things I did not want to do; (8) sent me nasty or hurtful messages online; (9) shared nasty or hurtful things about me online; (10) shared embarrassing photos of me online; and (11) threated me. The items were classified as belonging to four bullying types, namely, physical, verbal, relational, and cyber (see Table 1). The scaling system involved a frequentist four-point scaling as follows: never; a few times a year; once or twice a month; and at least once a week. The scaling was reversed in the present study so that high scores were indicative of enhanced bullying.

### 2.3. Data Analyses

#### 2.3.1. Multilevel Latent Class Analysis: Enumeration

Data were analyzed using a multilevel Latent Class (LC) mixture modeling approach to investigate population heterogeneity on the bullying latent trait [36,37,38,39,40,41] after accounting for school variability. In other words, we accounted for the nested structure of the data (students nested within schools) to adjust the errors of measurement for the non-independence of observations (termed autocorrelation, [42,43]). The goal of the model is to identify distinct latent subgroups using the information provided in the response vectors [44,45,46,47,48]. Each participant was assigned a probability of membership in all classes (summed to 100%) and the model engaged inferential statistics to optimally classify individuals into the best possible subgroup.

When subgroups became increasingly similar, entropy values, which reflect class homogeneity and latent class accuracy, decreased [49]. Entropy represents a weighted average of individuals’ posterior probabilities of membership [50], and the higher the probabilities, the more precise the assignment of persons to groups. However, despite its appeal as an index of class definition, it should not be valued towards concluding the optimal number of classes [49]. Initially, the latent class methodology involved fitting a one-class model to the data, which acted as a reference model; the data were then fitted to an increased number of subgroups until the model fit suggested otherwise. In the present study, we employed Mplus 8.9 and the mixture facility to identify an optimal solution among a range of one to six latent subgroups [51]. A model was properly run when the loglikelihood was replicated several times and using different start values (n = 20). The estimator was Maximum Likelihood with robust standard errors (MLR) to account for the categorical nature of the data. Several inferential statistical criteria and information indices were employed to conclude an optimal number of subgroups [52]. Information criteria involved the AIC, the BIC, the consistent AIC (CAIC, [53]), the Bayes factor, the correct model probability index of superiority (cmPk), the Schwartz criterion, and the approximate weight of evidence (AWE, [54]) criterion. The Akaike Information Criterion (AIC) engages estimates of the loglikelihood and the number of estimated parameters in the following manner:(1)AIC=2k−2ln⁡(LL)

AIC was used primarily for historical reasons as its main criticism has been the inflation of the number of subgroups, thus, as an index, it lacks parsimony [55]. The Bayesian Information Criterion (BIC, [11]) is estimated as follows:(2)BIC=−2LL+dlog⁡(n)

The consistent AIC uses the following formula:(3)$$CAIC=-2LL+[d\log\left(n\right)+1]$$

The Approximate Weight of Evidence criterion (AWE) uses the following formula:(4)$$AWE=-2LL+2d[\log\left(n\right)+1.5]$$

Besides utilizing information criteria, additional quantitative criteria were employed in the form of the approximate Bayes Factor (BF), which tests the relative fit between two models as follows [49]:BF_A,B_ = exp[SIC_A_ − SIC_B_](5)

SIC refers to the Schwarz Information Criterion (SIC; [11]), which is estimated as shown below:SIC = −0.5 × BIC(6)

Based on earlier work [52], estimates over 10 units on the BF factor suggest that there is strong evidence one model is superior compared with a competing model. Using a similar logic, the approximate correct model probability index (cmP) compares all models with the sum value being 1, assuming one of the tested models is the true model, among several competing models. It is estimated as follows:(7)$$cmP_{A}=\frac{exp({SIC}_{A}-{SIC}_{max})}{\sum_{j=1}^{J}exp({SIC}_{j}-{SIC}_{max})}$$

With SIC max being the maximum SIC score of Model *j* under scrutiny. Statistical criteria favoring one model over another involved a likelihood ratio test based on the unbiased bootstrap distribution. In addition to statistical criteria, optimal class solutions should have both theoretical and practical value. Interpretational ease and conceptual clarity are valued heavily when concluding the optimal solution, as statistical criteria alone can be biased for statistical reasons (e.g., by capitalizing on chance, reflecting sample idiosyncrasies, and reflecting low or enhanced power effects). Given the large sample size in the present study, the level of significance was set to 1% as a means to adjust for excessive levels of power.

#### 2.3.2. Multilevel Latent Class Analysis: Class Separation

Three indices were used for analysis, entropy, the average posterior probabilities (AvePP), and the odds of correct classification (OCC) [49,56]. Entropy estimates of 0.80 and above were indicative of good separation. The AvePP index estimated how well a model classified participants in their most likely class, thus it reflected classification uncertainty for each one of the classes with the maximum value being 1. Estimates greater than 0.70 signaled good separation. Last, the Odds of Correct Classification (OCC) is the ratio of AvePP to the estimated class membership proportions, thus it is another summary index of the classification accuracy (for an extended discussion, see [49]). Estimates greater than 5 signalled good separation, but also class accuracy [49,56].

#### 2.3.3. Power Analysis

A power analysis was conducted to address the presence of at least six distinct subgroups (latent classes) with 11 indicators each. Subgroups were posited to have −2, −1, 0, 1, 1.5, and 2 mean values in logits across the 11 predictors, suggesting adequate differentiation between subgroups. A Monte Carlo simulation with n = 3500 participants and 6 classes was run using Mplus 8.8. Coverage across 1000 replicated samples ranged between 79.2% and 90.2%. The omnibus chi-square test was well-powered with the observed number of rejections being 5.3% given an expected number of rejected models at the nominal level of 5%. Thus, the current sample of 3874 participants sufficed to properly define and differentiate five distinct classes of individuals based on their bullying behavioral patterns. Recent recommendations suggested that sample sizes above *n* > 500 [57] are necessary for identifying the presence of latent subgroups (see also [58]).

## 3. Results

### 3.1. Latent Class Enumeration of Bullying Experiences

A series of latent class models were run (one to six classes) to identify an optimal number of subgroups that best described the latent construct of bullying. As shown in Table 2, among the information criteria, a five-class solution was favored by the BIC, CAIC, SIC, Bayes factor, and cmP for model ‘k’. As expected, the AIC favored more subgroups and so did AWE, for which little research is available on its behavior. Additional concluding evidence came from contrasting nested models using the Vuong−Lo−Mendell-Rubin LRT, and the adjusted Lo−Mendell−Rubin LRT; all of the tests favored a five-class solution over a four-class proposition (Lo−Mendell−Rubin LRT(12) = 186.726, *p* < 0.001; Adjusted Lo−Mendell−Rubin LRT(12) = 184.0861, *p* < 0.001). Furthermore, all of the tests were non-significant when contrasting a six-class model to a five-class solution. Consequently, collectively, all of the results corroborated with a preference for the presence of five distinct subgroups (see Figure 1) with ample separation (entropy = 0.790) and adequate sample sizes.

### 3.2. Latent Class Separation

One of the most important conditions that points to the preference and selection of the most optimal solution relates to class separation, was demonstrated with significantly different levels in the indicators across classes [59]. Several indicators pointed to good class separation. First, the entropy estimate for the five--class model was equal to 0.790. Second, the average posterior probability of classification (AvePP) was equal to 0.83 (greater than a cutoff of 0.70, [56]) and the Odds of Correct Classification (OCC) were 10.13 (greater than cutoff estimates of 5). Thus, collectively, evidence of proper levels of class separation was provided.

### 3.3. Latent Class Interpretation of Bullying Experiences

A very important criterion in the enumeration process is latent class interpretation. Class 1 represented a student group with very low to non-existent experiences of bullying (low bullying). On the opposite pole, there was a much smaller group of students who experienced the maximum number of bullying experiences across all four types (high bullying). A mid-experiences group (class 3) was observed with average experiences across all types (average bullying), with, interestingly, the second highest rates of cyberbullying experiences. The last two classes, namely, classes 2 and 4, had nonexistent cyberbullying experiences and above average (for class 4) or below average (for class 2) experiences of physical, verbal, and relational bullying (termed above average P-V-R and below average P-V-R).

### 3.4. Bullying Experiences and Gender Effects

Table 3 shows the role of gender in bullying with the distribution of males and females differing across classes based on the omnibus chi-square test [*χ*^2^(4) = 262.268, *p* < 0.001]. As shown in Table 3, when viewing the most extreme classes, there were significantly more females in Class 1 (67%), the no-bullying class, compared with males (33%) and significantly more males (71%) in the extreme bullying class (i.e., class 5) compared with females (29%). No differences were evident in class 2, the low bullying class for which male and female ratios were around 50%. For the medium bullying class (class 3), again, the prevalence of males was elevated (63%) compared with the presence of females (37%). Lastly, class 4, defined by a high frequency of mild bullying symptoms and a low frequency of serious bullying behaviors, was again favored by males (64%) compared with females (36%). Overall, the frequency of male presence was increased, as was the number of bullying behaviors.

A gender differences analysis at the item level was conducted by fitting a latent class model to the data using the “knownclass” approach, with gender as the independent variable (see Figure 2). The goal of this analysis was to identify gender differences across bullying behaviors in terms of prevalence. Table 4 shows odds ratio statistics and their corresponding significance. As shown in the table, bullying experiences were more prevalent in males across all behaviors and types with a two-fold to three--fold increase in these experiences for males compared with females. Although these effects reflected small to medium effect sizes [60], they were nevertheless significantly different from zero.

## 4. Discussion

The purpose of the present study was to profile bullying behaviors in elementary schools in Saudi Arabia. A secondary purpose was to examine differences in bullying behaviors across gender. Several important findings emerged and are discussed in order of importance.

The most important finding was that five distinct profiles of bullying were observed, with differences in level and content. Besides being very low and very high in bullying classes (i.e., classes 1 and 5), three new, qualitatively different subgroups emerged (classes 2, 3, and 4). Class 3 appeared to be a medium leveled class, but what was striking about this class is that the levels were approximately equal across all four domains of bullying. This was notable, despite observing two emergent classes for which cyber levels were non-existent, despite above average (for class 4) or below average (class 2) levels of physical, verbal, and relational types of bullying experiences. These findings had both similarities and differences with past research. For example, low and high bullying experiences groups have been similarly observed in past research (see [61,62] reported) that up to 11% of the student population exhibited all types of bullying behavior. In the present study, this was reflective of the sum of the participants in classes 3 and 5. These two classes amounted to 501 students, which, in percentage, reflected 12.9% of the student population of fourth graders who participated in TIMSS in Saudi Arabia, and thus this finding has a striking resemblance to past research. Any bullying experiences are reported in around 60% of the student population in middle school [63] or 41% [64]. In the present study, summing up the participants in all classes except class 1 amounted to 44% of the national sample, much lower compared with [65], but similar to [64]. Furthermore, the present low levels of cyberbullying reported herein agree with past research where cyberbullying behaviors had the lowest frequency [66], but disagree with the findings by [31], in that between 1990 and 2009 cyber bullying levels increased. Interestingly, levels of cyber bullying are expected to increase by adolescence [67]. The present findings also disagreed with the profiling by [68], who reported lower levels of verbal aggression in fifth graders compared with the ones observed in the present study.

Another important finding was that gender significantly differentiated the profiles. For example, for the anchor profiles, low and high bullying were predominantly female and male, respectively. Thus, there was evidence of significant differences in the level of bullying across gender. Differences in content were also observed; for example, Class 3 had high levels of physical aggression (e.g., damaging property, forcing, and stealing) and this class was comprised of mostly males (63% vs. 37). This finding agreed with past studies, pointing to the high prevalence of aggressive incidences and perpetrator behaviors by boys compared with girls [61,69,70,71]. Item-level gender differences pointed to significantly elevated experiences of bullying for males compared with females across the board of bullying behaviors and types. This finding disagreed with past studies that reported a higher prevalence of relational bullying for females [72,73] compared with males; however, this difference could be a function of measurement differences or age. In the present study, relational bullying was assessed with a single item. Furthermore, there has been evidence that gender differences in bullying are moderated by age levels [27,28] or context [4,74,75], thus complicating the picture on the causes behind the observed differences in bullying across gender. The present study also has important implications for educational policy, and these are discussed next.

Last, the profiles that emerged in the present study deviated markedly from those observed in past studies using the latent class methodology. To the best of our knowledge, no study has reported the existence of five profiles in the elementary school [67], reported five classes in middle and high schools); instead, the number of emerged subgroups ranged between three [32,76,77] and four in most studies [64,78,79].

### 4.1. Conclusions

Several conclusions were drawn from the present study. First, differences between males and females were predominantly in level across all types of bullying behaviors, but also a preference for boys to engage more with physical types of bullying. Second, cyber-bullying incidents were relatively low, being present in a small percentage of students. Third, the emerged profiles strikingly resembled those from past research and Western countries, pointing to similar international trends and effects.

### 4.2. Implications for Educational Policy and Practice

Bullying and victimization behaviors are a threat to the school’s safety and require immediate action. Utilizing a model from identification to intervention, students and school staff need to be cognizant of and recognize all behaviors related to bullying. The present results define and inform risk profiles for which immediate attention is required, particularly in the early stages of educational reform, before levels of bullying reach higher proportions. For example, 4% of the fourth graders reported extremely elevated bullying behaviors across all items. Another 6.5% had a high prevalence of bullying, except for online and cyberbullying. These two subgroups are certainly candidates for focused interventions on physical forms of aggression using targeted interventions from social workers, advisory groups, counselors, psychologists, focus groups, and policymakers.

Subsequent procedures need to be put in place following identification, such as the delineation of clear policies that prohibit bullying. This proposition comes in light of earlier findings that relational bullying observed in girls has been given little attention by school staff [80]. Support for students can take the form of providing counseling services to both bullies and victims. Creating peer support groups can also prove to be extremely beneficial. Last, engaging parents in anti-bullying campaigns and involving them in the school culture can also provide another layer of support.

### 4.3. Limitations and Future Directions

The present study is limited for several reasons. First, data came from the 2019 cohort, which reflected a 4-year lag from the present day. Second, the stability of the observed subgroups needs to be verified in future cross-sectional and longitudinal studies (see [12]). Third, the current instrument was rather comprehensive, and a more expanded profile of bullying behaviors could not be ascertained. Last, the present study followed a quantitative protocol that evaluated magnitude and quantity rather than quality; thus, qualitative methodologies may further contribute to our understanding of the “whys” of bullying behaviors (see [81]). In the future, it will be interesting to evaluate proximal and distant factors that are predictive of latent class membership. For example, by applying an ecological framework, one could answer the question “What are the individual proclivities (student level), family environment characteristics (home), and school-related experiences (school environment) that are predictive of membership in specific bullying profiles?”. Applications of problem behavior theory, social dominance theory, stress and coping theory, and strain theory may also be successful when exploring pathways to bullying behaviors [73,82]. Last, socioeconomic factors exert salient effects on bullying and need to be accounted for in future studies [76].

## Figures and Tables

**Figure 1 children-10-00610-f001:**
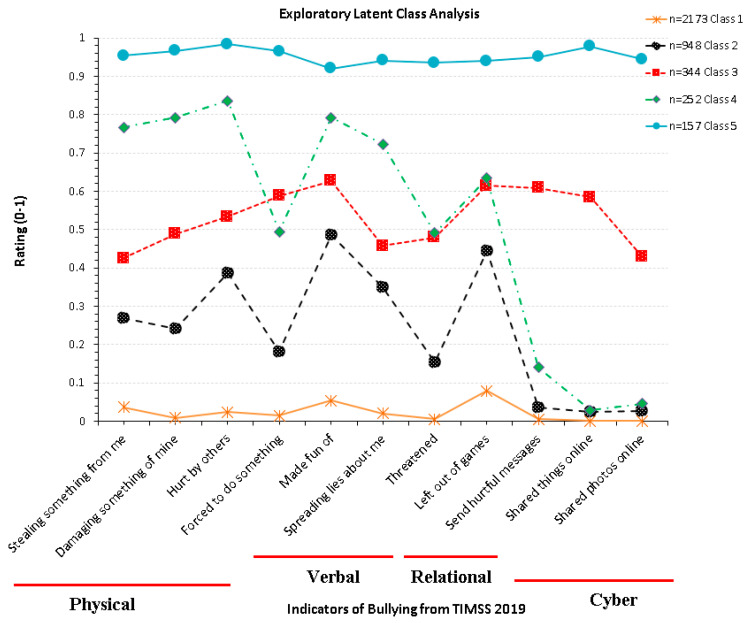
Optimal latent class solution when profiling bullying behaviors on TIMSS.

**Figure 2 children-10-00610-f002:**
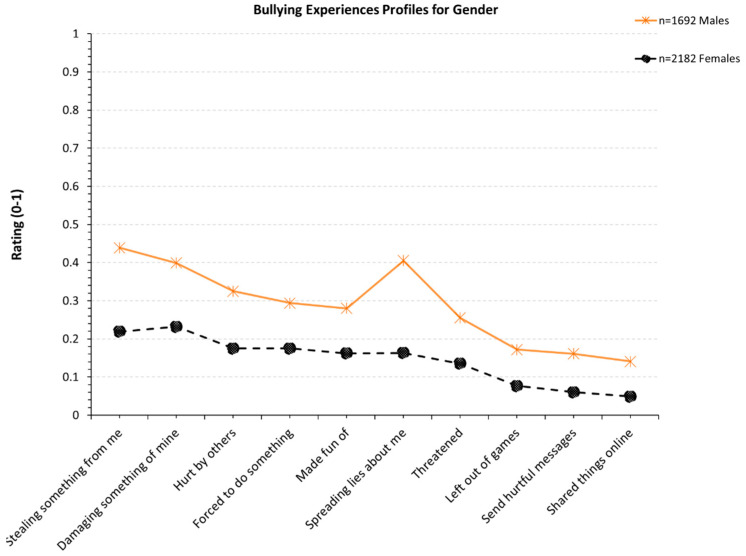
Bullying experiences across gender.

**Table 1 children-10-00610-t001:** Items and types of bullying in the TIMSS 2019 Scale.

Items of Bullying Scale of TIMSS 2019	Type of Bullying Behavior
1. Stealing something from me	Physical
2. Damaging something of mine	Physical
3. Hurt by others	Physical
4. Forced to do something	Physical
5. Made fun of	Verbal
6. Spreading lies about me	Verbal
7. Threatened	Verbal
8. Left out of games	Relational
9. Send hurtful messages	Cyber
10. Shared things online	Cyber
11. Shared photos online	Cyber

**Table 2 children-10-00610-t002:** Model fit for 1–6 classes when profiling bullying experiences.

Model Tested	LL	Npar	AIC	BIC	CAIC	AWE	Bayes Factor	cmP(k)	SIC	Entropy
1-class	−20,729.39	11	41,480.79	41,549.67	41,560.67	41,673.55	0.000	0.000	−20,774.8	-
2-class	−16,576.91	23	33,199.82	33,343.84	33,366.84	33,602.87	0.000	0.000	−16,671.9	0.875
3-class	−15,848.21	35	31,766.43	31,985.60	32,020.60	32,379.77	0.000	0.000	−15,992.8	0.869
4-class	−15,690.87	47	31,475.75	31,770.06	31,817.06	32,299.38	0.000	0.000	−15,885.0	0.829
5-class	−15,597.47	59	31,312.94	31,682.40	31,741.40	32,346.86	>100	1.000	−15,841.2	0.790
6-class	−15,565.29	71	31,272.58	31,717.19	31,788.19	32,516.79	0.000	0.000	−15,858.6	0.797

Note: LL = loglikelihood; Npar = number of estimated parameters; AIC = Akaike Information Criterion; BIC = Bayesian Information Criterion; CAIC = consistent AIC; AWE = approximate weight of evidence criterion; BF = Bayes factor; cmP(k)_ = correct model probability; SIC = Schwartz information criterion; LRTS = −2(LL_0_ − LL_1_); dtsc: (p_0_ * c_0_ − p_1_ * c_1_)/(p_0_ − p_1_); Chi-square = LRTS/dtsc; d.f. = p_1_ − p_0_.

**Table 3 children-10-00610-t003:** Roles of auxiliary variables in the latent class composition in the optimal five-class model.

Covariates	Class 1	Class 2	Class3	Class 4	Class 5
Males	33.1%	51.0%	62.8%	63.8%	71.4%
Females	66.9%	49.0%	37.2%	36.2%	28.6%

Note: The ratio of male to female participants was significantly different across all classes. Percentages refer to column percentages in that the total frequencies across gender per class are equal to 100%.

**Table 4 children-10-00610-t004:** Contrasts between males and females on bullying items.

Items of Bullying Scale	Odds Ratio	S.E.	Lower 95% C. I.	Upper 95% C. I.
1. Stealing something from me	2.791 *	0.292	2.274	3.426
2. Damaging something of mine	2.193 *	0.224	1.794	2.679
3. Hurt by others	2.277 *	0.257	1.825	2.840
4. Forced to do something	1.972 *	0.228	1.572	2.472
5. Made fun of	2.010 *	0.223	1.617	2.499
6. Spreading lies about me	3.498 *	0.377	2.832	4.320
7. Threatened	2.186 *	0.244	1.757	2.720
8. Left out of games	2.490 *	0.315	1.943	3.191
9. Send hurtful messages	3.024 *	0.433	2.283	4.005
10. Shared things online	3.174 *	0.480	2.360	4.269
11. Shared photos online	2.390 *	0.280	1.900	3.007

Note: Odds ratio (O.R.) values reflect the odds for males compared with females. Thus, an OR greater than 1 is indicative of a higher prevalence of bullying behaviors for males. * *p* < 0.05. S.E. = Standard Error. C.I. = Confidence Interval.

## Data Availability

Data are available from the official study of TIMSS 2019.

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
