# Peer review of "Profiling Experiences of Bullying in the Elementary School: The Role of Gender"

_children, 2023, doi:10.3390/children10040610_

Round 1

Reviewer 1 Report

Thank you.

I would like to strongly encourage authors to reformulate their manuscript with the changes.

Profiling Experiences of Bullying in the Elementary School: The Role of Gender (children-2303723)

Initial comment: In general, the manuscript seems to me an appropiate paper to be published. The authors made a sufficiently consistent introduction and the analysis methods are adequate. The discussion responded to a greater or lesser extent to the research problems.

However, I would like to make some clarifications that would improve the quality of the article.

Please answer each of the sections separately:

Abstract: The abstract is ok according to the journal´s guidlines, I recommend reducing the number of keywords.

1. Introduction: The introduction is appropriate and summarizes the objective of the study. However, I would expand the introduction a bit with current studies. At the same time, the hypotheses or research problems must be established at the end of the section.

2. Material and methods:

Participants: It must be indicated how the sample was selected for the investigation. Informed consent must be mentioned.

“Procedures”: The procedure performed must be indicated, nothing is mentioned.

A detailed description of the questionnaires as well as Cronbach's Alpha should be mentioned.

Measure: Statistical analysis should be written more clearly relating it to the research objectives or intended results.

3. Results: I would introduce some causal relationship in the form of a structural equation model for contrast models that propose causal relationships between the study variables.

4. Discussion: The discussion is correct according to the information provided, although I still think that a better statistical treatment would considerably improve the manuscript.

5. Conclusions: It is necessary to introduce the “conclusions” section. The sections "Limitations of the study" and "Future prospects" are already included.

Minor revisions:

a)      Alpha Cronbach must be indicated in the measurements.

b)      Revise the whole bibliography

Final comment: I would like to strongly encourage authors to reformulate their manuscript with the changes made in this document.

Thank you very much.

Author Response

Thank you for your thoughtful review. Please let us know if you need additional information.

Profiling Experiences of Bullying in the Elementary School: The Role of Gender (children-2303723)

Initial comment: In general, the manuscript seems to me an appropiate paper to be published. The authors made a sufficiently consistent introduction and the analysis methods are adequate. The discussion responded to a greater or lesser extent to the research problems.

However, I would like to make some clarifications that would improve the quality of the article.

Please answer each of the sections separately:

Abstract: The abstract is ok according to the journal´s guidelines, I recommend reducing the number of keywords.

Answer:

Thank you, we have reduced the number of keywords as suggested.

  1. Introduction: The introduction is appropriate and summarizes the objective of the study. However, I would expand the introduction a bit with current studies. At the same time, the hypotheses or research problems must be established at the end of the section.

Answer:

Thank you, we included a best synthesis of studies as there are way too many and we had to balance both what is considered as “classic” studies and also most recent findings. We have updated this review as suggested with several more recent entries.

Non directional research questions were also stated as we were reluctant to provide specific expectations as some readers may consider expectations as a form of biased intent. The research questions were the following:

  1. What is the composition and number of latent subgroups related to bullying behaviors?
  2. Are there differences across gender in level and type of bullying behaviors?

  1. Material and methods:

Participants: It must be indicated how the sample was selected for the investigation. Informed consent must be mentioned.

Answer:

Thank you, but these data came out of one of the most population international databases that is the TIMSS. Thus, all ethical and related procedures for data collection have been monitored closely by the organization who committed the data collection. Thus, given that the data are available through a repository, we do not have access to informed consent forms, but the TIMSS team has certainly accommodated those processes as they follow the Helsinki Declaration for ethical data collection. For sample selection we have added the following information in the second paragraph of the “Participants and Procedures” section as follows:

“Participants were selected using a stratified two-stage cluster sample design. The first stage involved sampling schools with selection probabilities proportional to their size. Stratification involved region and geographical area, SES, and school type (private or public). The second stage involved selecting intact classes using a “within-scho0ol sampling” methodology developed by IEA Hamburg and Statistics Canada. The National Research Coordinators were then responsible for carrying out the assessments. Non response bias was accounted for by developing minimum inclusionary standards for schools (at least 85% participation).”  

“Procedures”: The procedure performed must be indicated, nothing is mentioned.

Answer:

Thank you, we have now added a section on procedures where we describe standardized practices that were implemented in the TIMSS. The new text reads as follows:

“The National School Coordinators arranged for the dates, times, and places of testing. They coordinated all procedures by collecting parent permission forms where necessary, the production and distribution of questionnaires and the completion of tracking forms. They were also responsible for securing assessment material and also arranging the return of completed forms the national center post administration.”

A detailed description of the questionnaires as well as Cronbach's Alpha should be mentioned.

Answer:

Thank you, we have now added information on Cronbach’s alpha with the estimate being 0.864, which is acceptable. As for the questionnaire, we present all items in Table 1, as well as the scaling in the methods section so we are unsure about what else to mention. Certainly, the reader who is interested about this specific instrument could read relevant reports from IEA and Boston College publications which came out of the group that handled the TIMSS data.

Measure: Statistical analysis should be written more clearly relating it to the research objectives or intended results.

Answer:

Thank you, we have made every effort possible to clarify the presented concepts and make them more accessible. This is not always easy and we tried to avoid being didactic as well. So, it was a critical balance but we tried hard to simplify our language.

  1. Results: I would introduce some causal relationship in the form of a structural equation model for contrast models that propose causal relationships between the study variables.

Answer:

Thank you, this is very interesting comment, although structural models are still based on correlations and covariances, thus, there is nothing really causal about them, unless they are engaging an experimental design. That was not the case in the present study. Also, we could not use structural equation modeling as our analytical approach was person-based rather than variable-based. Thus, our analytical approach targeted at grouping individuals (that is why we used LCA) and not variables (as in SEM). Hope that clarifies the issue.

  1. Discussion: The discussion is correct according to the information provided, although I still think that a better statistical treatment would considerably improve the manuscript.

Answer:

Thank you, we have tried to clarify difficult concepts and make relatively complex content applicable to a wider audience. Hopefully we are now conveying our data analytic methodology more efficiently.

  1. Conclusions: It is necessary to introduce the “conclusions” section. The sections "Limitations of the study" and "Future prospects" are already included.

Answer:

Thank you, we have now added a conclusions section with summary statements from the present study’s findings. This new section now reads as follows:

4.1 Conclusions

“Several conclusions are drawn from the present study. First, differences between males and females were predominantly in level across all types of bullying behaviors but also a preference for boys to engage more with physical types of bullying. Second, cyber-bullying incidents were relatively low by being present in a small percentage of students. Third, the emerged profiles strikingly resemble those from past research and western countries pointing to similar international trends and effects. “

Minor revisions:

  1. Alpha Cronbach must be indicated in the measurements.
  2. Revise the whole bibliography

Answer:

Thank you, there is ample evidence that Cronbach’s alpha is inferior in many respects compared to coefficient omega that we utilized. This is evident in our earlier publication in which the pros and cons of alpha are presented.

Sideridis, G. D., Saddaawi, A., & Al-Harbi K. (2018). Internal consistency reliability in measurement: Aggregate and multilevel approaches. Journal of Modern Applied Statistical Methods, 17(1), Article 15.  

We nevertheless added Cronbach’s alpha which in the present study was 0.864, which is adequate.

The bibliography was checked for accuracy and consistency to ensure that all references are in the main text and the opposite. We furthermore, added more recent references as suggested.

Final comment: I would like to strongly encourage authors to reformulate their manuscript with the changes made in this document.

Answer:

Thank you, we appreciate your thoughtful responses and effort that was put in your review.

Thank you very much.

Reviewer 2 Report

"Profiling Experiences of Bullying in the Elementary School: The Role of Gender":

Abstract:

Provide more background information: It would be helpful to provide some context and background information on the prevalence of bullying in elementary schools in Saudi Arabia, as well as the potential negative impacts of bullying on children's well-being.

Clarify the research questions: While the primary research question is clear, the secondary research question could be further clarified. Specifically, it would be helpful to specify what types of gender differences in bullying behaviors were examined.

Provide more detail on the methodology: The abstract could benefit from more information on the specific methods used to analyze the data, such as what software or statistical tests were employed.

Be more specific about the results: Instead of using general terms like "low," "medium," and "high," it would be more informative to provide specific numerical values or percentages for each profile of bullying experiences.

Discuss implications in more detail: The implications for educational policy could be further elaborated upon in the abstract, providing specific recommendations for how schools can address bullying behaviors and support the well-being of their students.

The Introduction of the paper provides a good overview of the problem of bullying in schools and its implications for children's well-being and academic achievement. However, there are a few ways in which it can be improved:

Provide more context: While the introduction mentions that bullying rates have increased in recent years, it doesn't provide any context for why this might be the case. It would be helpful to briefly discuss some of the social, cultural, and technological factors that may be contributing to the rise in bullying.

Clarify the focus of the study: While the introduction mentions that the study will profile bullying behaviors in elementary schools in Saudi Arabia and examine gender differences, it could be clearer about what specific research questions the study is addressing. For example, what are the main types of bullying behaviors observed in Saudi Arabian elementary schools, and how do these behaviors differ between boys and girls?

Define key terms: The introduction uses terms like "bullying," "person-centered approach," and "academic achievement" without providing clear definitions or explanations. It would be helpful to briefly define these terms or provide references for readers who may be unfamiliar with them.

Provide a stronger rationale for studying gender differences: The introduction notes that gender differences in bullying have been observed in previous research, but it doesn't explain why it is important to examine these differences specifically in the context of Saudi Arabian elementary schools. Providing a clearer rationale for this aspect of the study would help to contextualize the research question and make it more compelling to readers.

Overall, the Introduction provides a good starting point for the study, but could benefit from additional context, clarification, and explanation to better situate the research question and motivate the study's goals.

The Methods section is well-written and clear. However, a couple of suggestions for improvement could be:

It would be helpful to provide more details about the data collection procedure. For example, how were participants selected? Was there any sampling strategy used to ensure a representative sample of the population? Were any incentives offered to encourage participation?

The authors should provide more information about the statistical analyses used in the study, particularly regarding the multilevel latent class analysis. For instance, it would be helpful to provide more details about how the model was specified and estimated, including the priors used and convergence diagnostics. Additionally, it would be useful to describe the assumptions underlying the approach and how they were checked.

Results:

The section on latent class enumeration and separation seems well-explained and adequately supported by the analyses conducted. However, it may be helpful for the authors to provide more detailed explanations of the specific bullying behaviors that were included in each of the five identified classes. This could help readers better understand the differences between the classes and how they relate to the different types of bullying experiences.

In addition, it would be helpful for the authors to provide some additional information on the sample used in the study. For example, they could provide information on the demographic characteristics of the students (e.g., age, race/ethnicity, socioeconomic status) and the schools from which they were recruited (e.g., Public vs Private). This information could help readers better understand the generalizability of the study findings.

Finally, while the authors do discuss the role of gender in bullying experiences, it would be helpful for them to provide more detailed explanations of how gender may be related to the different types of bullying behaviors. For example, they could discuss possible explanations for why males may be more likely than females to engage in certain types of bullying behaviors, and vice versa.

Provide more detail on the statistical methods used to run the latent class models, including the software used and any additional assumptions or constraints imposed.

Clarify the meaning and interpretation of the entropy estimate, Average Posterior Probability of Classification, and Odds of Correct Classification. Provide more context and explain why these metrics are important for evaluating class separation.

Discussion:

suggestions for improvement in the Discussion section:

Provide a brief summary of the study's main findings: While the Discussion section does mention the important findings, it may be helpful to provide a brief summary of the study's main findings at the beginning of the section to give readers an idea of what to expect.

Discuss the implications of the findings: The Discussion section can be expanded to discuss the implications of the study's findings. For example, what do the findings suggest about the prevalence and nature of bullying in elementary schools in Saudi Arabia? How can educators and policymakers use these findings to improve bullying prevention and intervention programs? Are there any limitations to the study that need to be taken into account when interpreting the findings?

Compare the findings to previous research: While the Discussion section does mention some previous studies, it may be helpful to provide a more detailed comparison of the present findings to previous research. For example, how do the findings in this study compare to previous studies that have used similar methodologies? What are the similarities and differences in the prevalence and nature of bullying across different cultures and age groups?

Implications and limitations: some issues below that the researchers could think about and consider in the paper:

While acknowledging the limitations of using data from the 2019 cohort, it would be helpful to provide suggestions for how future research can address this limitation. One could suggest using more recent data or comparing findings across multiple cohorts.

To address the stability of observed subgroups, future research could use longitudinal designs to examine changes in bullying behaviors over time and explore the reasons behind these changes.

Author Response

Thank you for your thoughtful responses!

Comments and Suggestions for Authors

"Profiling Experiences of Bullying in the Elementary School: The Role of Gender":

Abstract:

Provide more background information: It would be helpful to provide some context and background information on the prevalence of bullying in elementary schools in Saudi Arabia, as well as the potential negative impacts of bullying on children's well-being.

Answer:

Thank you, I wish we could but we could not locate a single study in Saudi Arabia that evaluates bullying in the elementary school. Thus, any inference would be speculative and we felt uneasy to say more about the context in the absence of quantitative or qualitative data.

Clarify the research questions: While the primary research question is clear, the secondary research question could be further clarified. Specifically, it would be helpful to specify what types of gender differences in bullying behaviors were examined.

 Answer:

Thank you, we have now posited non-directional research questions so that we won’t be accused of biased expectations. The research questions now read as follows:

  1. What is the composition and number of latent subgroups related to bullying behaviors?
  2. Are there differences across gender in level and type of bullying behaviors?

As shown above, the type of gender differences is expressed in the form of “level” as this is a cross-sectional quantitative study, only.

Provide more detail on the methodology: The abstract could benefit from more information on the specific methods used to analyze the data, such as what software or statistical tests were employed.

 Answer:

Thank you, we now include the software used in the abstract (Mplus 8.9). The method is already mentioned, “latent class analysis”.

Be more specific about the results: Instead of using general terms like "low," "medium," and "high," it would be more informative to provide specific numerical values or percentages for each profile of bullying experiences.

 Answer:

Thank you for your nice comment. The values on the Y-axis are actually percentages (0-1 range) and thus, the low, medium, and high conventions reflect approximately zero, approximately 50%, and over 90% for most bullying behaviors. Since these profiles were consistent across behaviors, the easiest way to describe them was using these three labels that express level up to 100%.

Discuss implications in more detail: The implications for educational policy could be further elaborated upon in the abstract, providing specific recommendations for how schools can address bullying behaviors and support the well-being of their students.

 Answer:

Thank you, we have now adopted this recommendation by succinctly providing content in the abstract. This content reads as follows:

“Implications for educational policy can clearly direct the development of support groups and expert counseling for both bullies and victims, staff training for identification and course of action, and the development of standardized school policies when such incidences occur.”

The Introduction of the paper provides a good overview of the problem of bullying in schools and its implications for children's well-being and academic achievement. However, there are a few ways in which it can be improved:

Provide more context: While the introduction mentions that bullying rates have increased in recent years, it doesn't provide any context for why this might be the case. It would be helpful to briefly discuss some of the social, cultural, and technological factors that may be contributing to the rise in bullying.

Answer:

Thank you, we have now included a new paragraph that briefly describes factors. However, we did not want to expand on this because we did not deal with predictors of bullying (besides gender) and possibly causal factors. Thus, we believe we should keep this to the minimum. The new text reads as follows:

“While bullying rates are on the rise, researchers have attempted to elaborate on the causal mechanisms of bullying. To this end, they have identified as potential causal mechanisms, the increased use of social media making it easier to behave in that manner compared to face-to-face social experiences (Canty, Stubbe, Steers, & Collings, 2016), peer pressures (Zhang, Tang, Li, & Jia, 2022), abusive or neglectful family environment (Fraga, Soares, Soares-Peres & Barros, 2022), personal dispositions (Francioli et al., 2016), and cultural factors (Nguyen, Bradshaw, Townsend, & Bass, 2020). ”

Clarify the focus of the study: While the introduction mentions that the study will profile bullying behaviors in elementary schools in Saudi Arabia and examine gender differences, it could be clearer about what specific research questions the study is addressing. For example, what are the main types of bullying behaviors observed in Saudi Arabian elementary schools, and how do these behaviors differ between boys and girls?

Answer:

Thank you, we have now introduced research questions that make things clearer. However, as for past research, our rigorous database search did not reveal a single study that investigated school bullying in Saudi Arabia. Thus, we could not link past research to the currently investigated phenomenon.

Define key terms: The introduction uses terms like "bullying," "person-centered approach," and "academic achievement" without providing clear definitions or explanations. It would be helpful to briefly define these terms or provide references for readers who may be unfamiliar with them.

 Answer:

Thank you, we have defined and clarified all these terms, except academic achievement that we consider common knowledge and also for the fact that we did not manipulate achievement as a construct in the present study..

Provide a stronger rationale for studying gender differences: The introduction notes that gender differences in bullying have been observed in previous research, but it doesn't explain why it is important to examine these differences specifically in the context of Saudi Arabian elementary schools. Providing a clearer rationale for this aspect of the study would help to contextualize the research question and make it more compelling to readers.

 Answer:

Thank you, we now provide additional evidence on the differences in bullying across gender, especially the prevalence of physical bullying in males and cyber in females but, we could not substantiate those differences in the context of Saudi Arabia as we could not locate a single study that examined subtypes of bullying in the country and the roles of gender for that purpose.

Overall, the Introduction provides a good starting point for the study, but could benefit from additional context, clarification, and explanation to better situate the research question and motivate the study's goals.

Answer:

Thank you, we have revised the introduction section and added more pertinent research and more recent studies to strengthen the previous findings and make the case for the present study.

The Methods section is well-written and clear. However, a couple of suggestions for improvement could be:

It would be helpful to provide more details about the data collection procedure. For example, how were participants selected? Was there any sampling strategy used to ensure a representative sample of the population? Were any incentives offered to encourage participation?

Answer:

Thank you, we have now expanded our methodology to include more information about sampling and the procedures that took place in data collection.

The authors should provide more information about the statistical analyses used in the study, particularly regarding the multilevel latent class analysis. For instance, it would be helpful to provide more details about how the model was specified and estimated, including the priors used and convergence diagnostics. Additionally, it would be useful to describe the assumptions underlying the approach and how they were checked.

Answer:

Thank you, we now include relevant information. There is no information about priors because there was no Bayesian analysis involved. But we did mention information about using random starts, about replicating the loglikelihood, and about the specific estimator used (i.e., MLR). We believe we have expanded these sections to include all that information.

Results:

The section on latent class enumeration and separation seems well-explained and adequately supported by the analyses conducted. However, it may be helpful for the authors to provide more detailed explanations of the specific bullying behaviors that were included in each of the five identified classes. This could help readers better understand the differences between the classes and how they relate to the different types of bullying experiences.

Answer:

Thank you, these behaviors are shown in the figures and because the scaling is 0-1, they are in percentage form. Thus, any class can be evaluated by looking at specific bullying behaviors in the latent class figure. The same is true with gender differences and the known-class approach that was implemented to contrast gender on bullying behaviors.

In addition, it would be helpful for the authors to provide some additional information on the sample used in the study. For example, they could provide information on the demographic characteristics of the students (e.g., age, race/ethnicity, socioeconomic status) and the schools from which they were recruited (e.g., Public vs Private). This information could help readers better understand the generalizability of the study findings.

Answer:

Thank you, we have now included more demographic information about the participants and their schools.

Finally, while the authors do discuss the role of gender in bullying experiences, it would be helpful for them to provide more detailed explanations of how gender may be related to the different types of bullying behaviors. For example, they could discuss possible explanations for why males may be more likely than females to engage in certain types of bullying behaviors, and vice versa.

Answer:

Thank you, we felt somewhat uncomfortable to discuss possible explanations as those would be merely speculative and the present design did not provide us with additional information that would allow us to make more meaningful attributions. In other words we do not have any results that can help us draw conclusions about why males and females engage in different types of behaviors. A qualitative enquiry will aid that purpose and enlighten us on the differences across gender and potential causes and consequences.

Provide more detail on the statistical methods used to run the latent class models, including the software used and any additional assumptions or constraints imposed.

Answer:

Thank you, we have used Mplus 8.9 and the mixture facility of the software.  Additional information involve inclusion of the estimator, the replication of the loglikelihood, and the employment of different numbers of random starts. Thus, we have highly edited the data analysis section to make it more detailed but also easier to read.

Clarify the meaning and interpretation of the entropy estimate, Average Posterior Probability of Classification, and Odds of Correct Classification. Provide more context and explain why these metrics are important for evaluating class separation.

Answer:

Thank you, we have now added text to define entropy, AVEPP and OCC. We have revised the section on data analysis and also provide information in section 2.3.2.

Discussion:

suggestions for improvement in the Discussion section:

Provide a brief summary of the study's main findings: While the Discussion section does mention the important findings, it may be helpful to provide a brief summary of the study's main findings at the beginning of the section to give readers an idea of what to expect.

Answer:

Thank you, we have now added a conclusions section for that purpose.

Discuss the implications of the findings: The Discussion section can be expanded to discuss the implications of the study's findings. For example, what do the findings suggest about the prevalence and nature of bullying in elementary schools in Saudi Arabia? How can educators and policymakers use these findings to improve bullying prevention and intervention programs? Are there any limitations to the study that need to be taken into account when interpreting the findings?

Answer:

Thank you, we have specified all the limitations we could think of but mainly refer to the cross-sectional nature of the study and the correlational findings. We also present several useful implications as in our view, rates of bullying are elevated in Saudi Arabia and certain steps need to be taken so that both structures and supports will be available to both bullies and victims.

Compare the findings to previous research: While the Discussion section does mention some previous studies, it may be helpful to provide a more detailed comparison of the present findings to previous research. For example, how do the findings in this study compare to previous studies that have used similar methodologies? What are the similarities and differences in the prevalence and nature of bullying across different cultures and age groups?

Answer:

Thank you, the latent class model is relatively new and thus, studies that employ the methodology and are pertinent to this age group, we believe we identified all of them. There are certainly many more studies in older ages such as adolescence, adulthood, and the workplace, but, they were not really relevant to the present study as they reflect different populations.

Implications and limitations: some issues below that the researchers could think about and consider in the paper:

While acknowledging the limitations of using data from the 2019 cohort, it would be helpful to provide suggestions for how future research can address this limitation. One could suggest using more recent data or comparing findings across multiple cohorts.

Answer:

Thank you, this is very interesting. First, this is the latest TIMSS data available, so, up to date, there are no new data. Second, cohorts cannot be compared (such as 4th and 8th graders) because the bullying scale is different and the same is true of previous administrations of the scale in 2015 vs 2019. So, the present report reflects the latest knowledge on the subject matter given the latest international data available.

To address the stability of observed subgroups, future research could use longitudinal designs to examine changes in bullying behaviors over time and explore the reasons behind these changes.

Answer:

Thank you, this is true and we have added the premise of longitudinal studies in our presentation of future directions.

Round 2

Reviewer 1 Report

Thank you for the effort made. Go ahead.